# Cohort profile: the eLIXIR Partnership—a maternity–child data linkage for life course research in South London, UK

Lauren E Carson [ID],[1] Borscha Azmi,[1] Amelia Jewell [ID],[2] Clare L Taylor,[3] Angela Flynn,[4] Carolyn Gill,[4,5] Matthew Broadbent,[2] Louise Howard,[2,6] Robert Stewart,[1,2] Lucilla Poston,[4] on behalf of the eLIXIR Partnership

For numbered affiliations see end of article.

**Correspondence to**
Dr Lauren E Carson;
lauren.carson@kcl.ac.uk

## ABSTRACT

**Purpose** Linked maternity, neonatal and maternal mental health records were created to support research into the early life origins of physical and mental health, in mothers and children. The Early Life Cross Linkage in Research (eLIXIR) Partnership was developed in 2018, generating a repository of real-time, pseudonymised, structured data derived from the electronic health record systems of two acute and one Mental Health Care National Health Service (NHS) Provider in South London. We present early descriptive data for the linkage database and the robust data security and governance structures, and describe the intended expansion of the database from its original development. Additionally, we report details of the accompanying eLIXIR Research Tissue Bank of maternal and neonatal blood samples.

**Participants** Descriptive data were generated from the eLIXIR database from 1 October 2018 to 30 June 2019. Over 17 000 electronic patient records were included.

**Findings to date** 10 207 women accessed antenatal care from the 2 NHS maternity services, with 8405 deliveries (8772 infants). This diverse, inner-city maternity service population was born in over 170 countries with an ethnic profile of 46.1% white, 19.1% black, 7.0% Asian, 4.1% mixed and 4.1% other. Of the 10 207 women, 11.6% had a clinical record in mental health services with 3.0% being treated during their pregnancy. This first data extract included 947 infants treated in the neonatal intensive care unit, of whom 19.1% were postnatal transfers from external healthcare providers.

**Future plans** Electronic health records provide potentially transformative information for life course research, integrating physical and mental health disorders and outcomes in routine clinical care. The eLIXIR database will grow by ~14 000 new maternity cases annually, in addition to providing child follow-up data. Additional datasets will supplement the current linkage from other local and national resources, including primary care and hospital inpatient data for mothers and their children.

## Strengths and limitations of this study

► The Early Life Cross Linkage in Research (eLIXIR) is a unique population-based database incorporating clinical data from maternity, neonatal and mental health records enabling life course studies of physical and mental health in a large, diverse, inner-city, UK population.

► Studies undertaken using eLIXIR will have not only have implications for local healthcare improvement but also the potential to provide evidence to influence healthcare in similar national/global settings.

► Missingness and inaccuracy in all routine/administrative clinical databases will be a key limitation to this database.

► The representativeness of the cohort to the UK population is limited to mixed, inner-urban catchments.

in utero, through to infancy, childhood and into adulthood.[1 2] Much of the supporting evidence underpinning a 'life course approach' to disease prevention from pregnancy and infancy onwards has been accrued from large birth cohorts[3–6]; however, directly recruited cohorts are by definition drawn from individuals recruited over a prespecified and limited time period and thereby become rapidly outdated as temporal shifts in population demography, lifestyle and ethnicity occur. Moreover, facility-centred follow-up is expensive and difficult to sustain.[7 8] Sample attrition is common and can introduce significant methodological biases that may affect the validity of investigations into novel risk outcomes.[9] In addition, direct recruitment may result in cohorts with a limited representation of the target population because of selective inclusion. Population-based registries also offer insight into rarer diseases and outcomes, not feasible in current research cohorts. Linked administrative data are

## INTRODUCTION

Investment in health in the earliest stages of life is increasingly recognised as a means to improve the life course of health; beginning

increasingly used to provide evidence to guide policy and clinical management.[10–12] The longitudinal nature of such case registers, their size and coverage of defined populations provide an increasingly attractive alternative to the study of birth cohorts for defining the early life exposures that contribute to the population burden of physical and mental health disorders. These can provide longitudinal information on large numbers of women and children, as well as the potential for linkage with a widening portfolio of available local and national datasets to follow health from birth to adulthood (https://digital.nhs.uk/data-and-information/data-collections-and-data-sets/data-sets). Both inclusion and attrition bias of traditional birth cohorts can be overcome through routine comprehensive health records, as these can capture rich clinical data in a given population on all women receiving antenatal care and their infants.[13] Although well established in Scandinavian countries, national birth registries in the UK have not been widely used in linkage programmes using infant and childhood data, although population registry data from Scotland have for many years provided information on relationships between maternal and neonatal outcomes that has informed clinical guidelines in the UK and beyond.[14] Several linkages of clinical maternity and infant data have nonetheless shown the feasibility of the approach and usefulness, for example, in aligning hospital maternity data with national birth registration datasets, or birth registration datasets with Hospital Episode Statistics (HES), or using UK primary care pregnancy data to create a pregnancy register.[15–25]

It is well established that maternal physical and mental well-being in pregnancy and the postpartum period can strongly influence the neonatal outcome and the physical and mental health of the child.[26–29] To our knowledge, no clinical data linkages in maternity or neonatal services have to date incorporated clinical information from maternity, neonatal and mental health services into a single continuum to interrogate these associations at a population level. The Early Life Cross Linkage in Research (eLIXIR) Partnership has been developed to address these relationships from early pregnancy, the perinatal period and beyond into later life. Funded by the Medical Research Council (MRC) in 2017, the partnership is a multidisciplinary academic collaboration that aims to combine maternal, infant and child health data into a single resource to allow information from large numbers of mothers, babies and children to be investigated over an unlimited time period. The intention is to provide a naturally accumulating database to support investigations into associations between physical and mental health in mother and child.

The eLIXIR Partnership provides a mechanism through which research datasets can be linked to clinical records, under appropriate and approved levels of anonymity and data security. An added benefit is the potential to incorporate data from multiple sources, for example, health, environment, social and education. There are, however, important ethical and legal considerations, as well as technical security requirements, if linkages are to be performed between sources of routinely collected clinical data and exemption from individual consent to be permissible.[30]

Another aspect of the eLIXIR Partnership is the eLIXIR Research Tissue Bank established to link the routinely collected maternal and neonatal clinical data with biological samples. This has an advantage over static cohort studies by providing a 'dynamic' collection of samples, enabling the identification of population trends and influences of new clinical interventions. The provision of samples from women attending antenatal care will provide a unique biobank to address mechanisms of common and rarer complications in pregnancy and in neonatal life, and their consequences for the longer-term health of the mother and child. Common complications will include gestational diabetes, mental illness, prematurity and pre-eclampsia. Similarly, by the provision of samples from neonatal intensive care, eLIXIR will contribute to a better understanding of neonatal morbidity and mortality.

With records of over 14 000 individual births per year, eLIXIR has the potential to become one of the largest mother–infant–child datasets in Europe. This has been facilitated by King's Health Partners (KHP): an Academic Health Sciences Centre that brings together one academic institution (King's College London) and three National Health Service (NHS) Foundation Trusts (Guys and St Thomas' NHS Foundation Trust (GSTT), King's College Hospital NHS Foundation Trust (KCH) and South London and Maudsley NHS Foundation Trust (SLaM)).

This manuscript details the technical and procedural elements in place to safeguard the legal and ethical rights of service users during the development and use of the eLIXIR database and to present the demographic profile of the eLIXIR population. Both technical and procedural elements draw strongly on experience gained in setting up the Clinical Record Interactive Search (CRIS) data resource at the Maudsley National Institute for Health Research (NIHR) Biomedical Research Centre (BRC).[31–33]

### Benefits of the system

Large data-linkage platforms, such as that created by the eLIXIR Partnership, provide a unique data warehouse through which important epidemiological questions can be asked, in the case of eLIXIR, within a large and diverse inner-city population. The ability to conduct these linkages allows not only the collection of a wide range of longitudinal health and social data, but also the capacity to support life course data analysis. The potential benefits arising from the use of clinical record 'big data' have been widely reported, and research databases such as eLIXIR are likely to increase in number due to the powerful and cost-effective nature of this research method.[34 35] eLIXIR is one of the first longitudinal research databases, from early pregnancy onwards, using routinely collected clinical data from maternity, neonatal

and mental health services that do not rely on a recruited cohort of participants.

## COHORT DESCRIPTION

### Data sources

Maternity and neonatal data were obtained from GSTT and KCH, and mental health data from SLaM. GSTT provides a full range of hospital and community services for people in Lambeth, Southwark and Lewisham, as well as specialist care for patients from further afield including referrals for high-risk pregnancies and neonatal complications. Similarly, KCH serves the boroughs of Lambeth, Southwark and Lewisham, but also Bromley, with specialist services to patients across a wider catchment area, including referrals for obstetrics and fetal medicine. SLaM provides comprehensive mental health services to a geographic catchment of over 1.2 million residents in four south London boroughs, Croydon, Lambeth, Lewisham and Southwark, as well as some regional/national specialist mental health services.

### Maternity, birth and neonatal intensive care data

The BadgerNet Platform (CleverMed) for routine clinical data is used extensively across the UK to create electronic patient records that capture early pregnancy community-based events and hospital-based events for low-risk and high-risk pathways of care (BadgerNet Maternity), and neonatal intensive care, neonatal transport, paediatric intensive care, neurology referrals and adult intensive care data (BadgerNet Neonatal). Within GSTT and KCH, the BadgerNet platforms are used for recording maternal/infant personal data, demographics, clinical history, clinic data (maternity only) and hospital episode data. The BadgerNet System records clinical records on a Single Care Record system, which is nationally hosted.[36] Although feasible within the BadgerNet System, linkage between maternity and neonatal data is not routinely conducted.

### Mental health data

Clinical records have been fully electronic across all SLaM NHS Trust mental health services since April 2006, using the bespoke electronic Patient Journey System (ePJS) that incorporated legacy data from earlier service-specific electronic health records. The CRIS platform[33] was developed in 2007–2008 and consists of a series of data-processing pipelines that both structure and de-identify PJS fields, rendering pseudonymiseddata from the full clinical record available at the researcher interface, with search and database assembly functionality facilitated by a front end, designed for non-technical use. The de-identifying process and its effectiveness, including the masking of identifying information in open-text fields and the generation of a pseudonymised identifier (CRIS ID), have been previously described.[32] The wider patient-led oversight and security models for CRIS have not changed significantly since it was established.[31–33] Ethical approval

was obtained for CRIS as a pseudonymised database for secondary analysis (Oxford C Research Ethics Committee, reference 18/SC/0372). In terms of cohort coverage, all SLaM care (including diagnoses, medication and services provided) is represented on CRIS, including Improving Access to Psychological Therapies data (IAPT; a large primary care service providing short-term psychological therapies).

### Data-linkage hosting environment

Data for eLIXIR are managed and stored at the Clinical Data-Linkage Service (CDLS) at SLaM: an impartial trusted third-party service that provides researchers access to linked clinical data in accordance with the strict governance conditions and processes agreed with relevant data controllers. The CDLS is managed by a small, dedicated team of informaticians, IT and Information Governance (IG) professionals (https://www.maudsleybrc.nihr.ac.uk/facilities/clinical-record-interactive-search-cris/), and currently hosts a range of datasets already linked with the SLaM CRIS mental health case register (eg, HES, National Cancer Registry, ONS death certification, Lambeth DataNet primary care records and National Pupil Database). The backbone of the eLIXIR database consists of a 'master patient index' (figure 1) allowing data to be robustly linked within an appropriately secure environment according to data specifications.

The CDLS hosts both source data and the master patient index, on behalf of the eLIXIR Partnership, on a secure server within the NHS firewall with role-restricted access. The CDLS additionally provides a data extraction service that meets security requirements, creating bespoke datasets for approved research use. Such derived data are managed by the approved research team and are hosted at all times on a dedicated drive within the NHS for analysis in this domain using hosted software already available at SLaM.

Four distinct services are offered by the CDLS as the data processor for the eLIXIR data. First, CDLS provides advice on permissions, approvals and contracts. These include the consideration of academic, technical, legal and ethical requirements. Second, CDLS facilitates data linkages either within the CDLS safe haven or via a third party, coordinating the secure transfer of data. Third, the CDLS is responsible for the secure storage of linked data in accordance with predefined IG and security standards. Fourth, the CDLS, as the custodian for the linked data, prepares and extracts bespoke and prespecified databases for approved eLIXIR projects and provides these to researchers. Therefore, there is no direct access by researchers to the full linked data files, enhancing data protection and confidentiality.

### Data-linkage procedures and resources

The eLIXIR Partnership uses common identifiers (eg, hospital number, NHS number, name and date of birth) to link between maternity, neonatal and mental health clinical data, which is undertaken by CDLS staff, not

The Data Flow Process

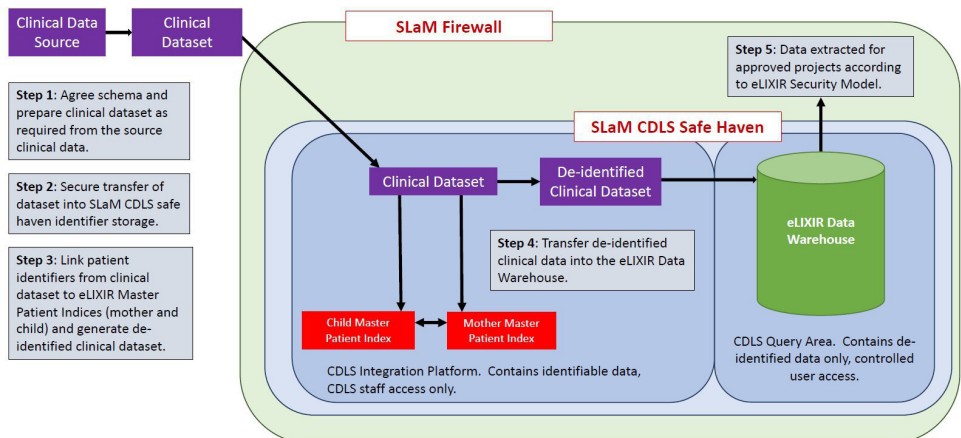

**Figure 1** Data flow diagram for the eLIXIR partnership linkage. CDLS, Clinical Data-Linkage Service; eLIXIR, Early Life Cross Linkage in Research; SLaM, South London and Maudsley National Health Service Foundation Trust.

researchers. Matching is undertaken using deterministic matching techniques on a given set of identifiers. Two records are said to match if all or some of the identifiers are identical, as defined by a hierarchical set of match-ranks. This creates a single master patient index including patient anonyms (mother and child), binary variables for presence/absence on each data source and key demographics. The source datasets and master patient index are stored by the CDLS behind the NHS firewall and linked data are extracted on a project-by-project basis by a CDLS informatician containing the variables and samples necessary for each study, with a study-specific encrypted anonym that contains completely pseudonymised data. In this way, databases are not stored in a linked format. The clinical data from maternity and neonatal services are extracted from BadgerNet Systems at both GSTT and KCH by information and communicaiton technology (ICT) staff at each site. These data are then sent securely to the staff at the SLaM CDLS and linked, using their common identifiers, with data from the CRIS system to incorporate mental health clinical data, where present, for each patient. This results in a comprehensive data resource to which researchers can apply for extracted data across maternity, neonatal and/or mental health services. Figure 1 details the dataflow for the eLIXIR Partnership database. Match quality is assured as 100% of infants born within the eLIXIR dataset were matched with their mothers' records with the BadgerNet System.

## IG framework

Results from linkages, current and prior, are stored within the CDLS safe haven, and the CDLS plays a key role in wider governance, supplementing the role of eLIXIR-specific oversight and data security, including the secure handling and storage of identifier fields required for data linkage. Section 251 (s251) of the NHS Act 2006 allows the common law duty of confidentiality to be set aside in specific circumstances where anonymised information is not sufficient and where 'opt-in' patient consent is not

practical. Opt-out information and details of the project are given to each patient entering maternity and neonatal services and patients have the option of opting out of the programme at any time. Approval under this legal framework was granted by the Health Research Authority (HRA) Confidentiality Advisory Group (CAG) to the eLIXIR team for all the above linkages, which allow data to be available in an identifiable format to a small number of data-processing staff in accordance with data sharing contracts between the data provider institutions (HRA CAG Ref: 18/CAG/0040). Therefore, for current and future data linkages within eLIXIR, ethical (REC) and s251 approval is required either through amendments to our existing agreements or new applications to these regulatory bodies. Activity for projects using linked datasets held by eLIXIR is audited by the eLIXIR Oversight Committee, helping to ensure that the researcher's project requirements (eg, clinical research, surveillance, service improvement or audit) are met and that projects progress within the agreed policy and practice framework. The primary role of the Committee is to provide the operational management of eLIXIR as identified in the eLIXIR Security Model and protocol for the eLIXIR Research Tissue Bank. In so doing, the Committee seeks to promote the scientific and ethical principles that should govern the use of eLIXIR data and seeks to represent stakeholders (who include the KHP Caldicott Guardians (individuals responsible for research governance at each NHS Trust site), service users, clinical professionals, lay persons and academics) and reflect their views and interests.

## Public and patient involvement

Public and patient involvement (PPI) involvement was incorporated throughout the development of the eLIXIR Partnership. The concept of the eLIXIR Partnership was presented to a variety of PPI groups, including the Maudsley BRC Data Linkage Service User and Carer Advisory Group,[37] Lambeth HealthWatch and the Young Persons Advisory Group at Great Ormond Street Hospital. PPI is ensured in the decision-making process of approving

eLIXIR projects through lay member representation on the eLIXIR Oversight Committee. The eLIXIR Oversight Committee reviews and approves all projects using eLIXIR data.

## Research Tissue Bank

The eLIXIR Research Tissue Bank is a prospective biobank of samples from pregnant women and infants being treated by GSTT. The Research Tissue Bank is integrated into the KCL Human Tissue Act (HTA) governance structure with active recruitment and collection of samples. All pregnant women aged over 16 years who are willing and able to give informed consent and infants admitted to the neonatal unit that have parental consent are eligible for inclusion in the eLIXIR Research Tissue Bank (Cambridge East Research Ethics Committee, reference 18/EE/0120).

### Maternal blood sample collection

Eligible women are recruited at the time of routine antenatal care venepuncture (11–15 weeks' gestation or later transfer of care or later antenatal care attendance) at GSTT. Women who agree to participate give written informed consent and an extra blood sample is collected at the same time as routine venepuncture, maximum volume 12 mL (2×6 mL tubes).

### Infant blood sample collection

It is our intention also to recruit samples from infants admitted to the neonatal intensive care unit (NICU), when blood is drawn for routine tests. Following written informed parental consent, residual blood from routine samples, which otherwise would be discarded, will be retained and collected. The samples will be processed and stored in a similar manner to the maternal blood samples.

Following birth in the community setting (home visits), every infant, whose mother has provided consent, is offered new-born blood spot screening to exclude metabolic disorders. A health professional pricks the baby's heel to collect four drops of blood on a card. For the biobank, an extra card to those taken for clinical purposes is used to collect extra bloodspots from the infant, after the routine spots are collected using the same heel prick. This sample is posted back to the eLIXIR team and stored in the research laboratory prior to transfer to a central storage facility.

All samples are transferred for processing in the research laboratory. After processing, all tubes are labelled with a study-specific barcode and entered on to a study-specific database (FreezerPro). The samples are stored in −80°C freezers for short term, before being transported to a central storage facility (NIHR BioResource, Milton Keynes).

## FINDINGS TO DATE
## Maternal, birth and birth outcomes

From the first data extraction (1 October 2018–30 June 2019) 10 207 women accessed antenatal care through GSTT or KCH maternity services with 8405 deliveries (8772 infants). This diverse, inner-city antenatal population was born in over 170 countries with a heterogeneous distribution across ethnic groups (table 1). Most were born outside the UK but most reported English as their primary language. Women were booked on average at 11.6 weeks gestation, which is slightly higher than the national guidelines of before 10 weeks.[38] The most common physical conditions experienced were; gynaecological problems (14%), asthma (8%) and pre-existing diabetes (6%). In addition, around one in five reported mental health problems, 3% were recorded as being exposed to female genital mutilation and 4% reported being current smokers at the time of their antenatal appointment (table 1).

With regard to birth episodes, twice as many women gave birth at GSTT than KCH with a mean gestational age at delivery of 38.8 weeks, and with 8% born prematurely (<37 completed weeks of gestation) and 3% born very prematurely (<34 weeks' gestation). Of the 8405 births, 8051 were singletons, 341 were twin births and 13 were triplets. Around half of deliveries were spontaneous cephalic, around one in five were emergency (or unspecified) caesarean section, and 15% elective caesarean section. Rates of stillbirth and neonatal deaths were 0.6% and 0.4%, respectively. The mean birth weight was 3257 g with 17% small for gestational age and 7% large for gestational age (table 2).

## NICU admissions

Of the 947 infants that had been treated in NICUs across both GSTT and KCH, 19.1% were postnatal transfers from an external trust and 8.7% were in utero transfers. The main reason for admission was for respiratory disease (28%) followed by preterm birth (22%). The average length of time spent in the NICU was 15.8 days. Of outcomes following admission 46% were readmitted to a postnatal ward, 29% were discharged home and neonatal death occurred in 4%, the remainder being unknown from the data available (table 3).

## Mental health

Of the 10 207 women attending antenatal care as registered in the eLIXIR database, 1184 had a clinical record in secondary mental health services (SLaM) with 307 women actively being treated at the time of their pregnancy (201 under the care of IAPT) (table 4).

## Research Tissue Bank

Following all necessary governance agreements samples were collected over a period of 3 months. A total of 1271 aliquoted samples (including EDTA, serum and whole blood) from 123 women were stored in the FreezerPro system. In this period, 63.4% of women approached gave consent to take part.

## Strengths and limitations

The eLIXIR Partnership has developed a unique population-based database incorporating clinical data from maternity, neonatal and mental health records.

**Table 1** Characteristics of women attending antenatal booking appointments at Guy's & St. Thomas' NHS Foundation trust (GSTT) and King's College Hospital NHS Foundation Trust (KCH) between first October 2018 and 30th June 2019

| | GSTT (n=5700) | KCH (n=4507) | Across both NHS trusts (n=10 207) |
|---|---|---|---|
| Age of mother at booking (years), mean±SD | 32.7±5.5 | 32.7±5.6 | 32.7±5.5 |
| Ethnicity, n (%) | | | |
| White | 2560 (44.9) | 2151 (47.7) | 4711 (46.1) |
| Black | 862 (15.0) | 1089 (24.3) | 1951 (19.1) |
| Asian | 474 (8.3) | 248 (5.6) | 722 (7.0) |
| Mixed | 218 (3.8) | 192 (4.3) | 410 (4.1) |
| Other | 248 (4.4) | 253 (5.6) | 501 (4.9) |
| Not recorded | 1338 (23.4) | 574 (12.7) | 1912 (18.8) |
| Born in the UK, n (%) | 2150 (37.7) | 2358 (52.3) | 4508 (44.1) |
| Primary language English, n (%) | 3711 (65.1) | 3474 (77.1) | 7185 (70.4) |
| Gestational age at booking (weeks), mean±SD | 11.9±6.4 | 11.2±6.4 | 11.6±6.5 |
| Parity, mean±SD | 0.7±1.1 | 0.8±1.1 | 0.8±1.1 |
| Gravida, mean±SD | 2.2±1.5 | 2.4±1.6 | 2.3±1.6 |
| Prior medical conditions, n (%) | | | |
| Haematological disorder | 231 (4.1) | 407 (9.0) | 638 (6.3) |
| Thrombosis | 57 (1.0) | 35 (0.8) | 90 (0.9) |
| Cardiac disorders | 103 (1.8) | 103 (2.3) | 206 (2.0) |
| Hypertension | 108 (2.0) | 85 (1.9) | 193 (1.9) |
| Renal disorders | 155 (2.8) | 135 (3.0) | 290 (2.9) |
| Asthma | 401 (7.0) | 413 (9.3) | 814 (8.0) |
| Lung disorders | 14 (0.2) | 10 (0.3) | 24 (0.2) |
| Diabetes | 375 (6.6) | 257 (5.7) | 627 (6.2) |
| Endocrine disorders | 307 (5.4) | 205 (4.6) | 512 (5.1) |
| Autoimmune disorders | 83 (1.4) | 62 (1.4) | 145 (1.4) |
| Epilepsy | 43 (0.7) | 37 (0.8) | 80 (0.8) |
| Neurological disorders | 146 (2.6) | 355 (7.9) | 501 (5.0) |
| Gastrointestinal disorders | 265 (4.6) | 282 (6.2) | 547 (5.4) |
| Gynaecological disorders | 742 (13.0) | 673 (15.0) | 1415 (14.1) |
| Liver disorders | 46 (0.9) | 55 (1.2) | 101 (1.0) |
| Bone disorders | 38 (0.7) | 59 (1.4) | 97 (0.9) |
| Joint disorders | 98 (1.7) | 77 (1.7) | 175 (1.8) |
| Back problems | 173 (3.0) | 375 (8.4) | 548 (5.1) |
| Genetic disorders | ND* | ND* | ND* |
| Congenital abnormalities | 29 (0.5) | 27 (0.7) | 56 (0.5) |
| Other | 85 (1.5) | 121 (2.6) | 206 (2.0) |
| Mental health problems (current or historical; self-reported), n (%) | 988 (13.8) | 1147 (25.5) | 2135 (20.9) |

Continued

**Table 1** Continued

| | GSTT (n=5700) | KCH (n=4507) | Across both NHS trusts (n=10 207) |
|---|---|---|---|
| Victim of female genital mutilation, n (%yes) | 209 (3.7) | 137 (3.0) | 346 (3.4) |
| Smoker at booking, n (%yes) | 195 (3.4) | 199 (4.4) | 394 (3.9) |

*Not disclosed (ND). Actual numbers suppressed to reduce risk of statistical disclosure.
NHS, National Health Service.

This is supplemented by the eLIXIR Research Tissue Bank. Together this resource will provide the basis for additional linkages to enable life course studies of physical and mental health in a large, diverse, inner-city, UK population. This will have not only implications for local healthcare improvement but also the potential to provide evidence to influence healthcare in similar national/global settings.

A limitation common to all routine/administrative clinical databases is source data missingness and inaccuracy. This, in contrast, is an advantage of prospective research cohorts. Nonetheless eLIXIR provides an opportunity for continuous feedback to the clinical provider on data absence or error. With regular meetings, we already appraise the Trust IT teams of, for example, duplicate patient data entry and missing data, especially that required for the national Maternity Services Data Set. Thus, eLIXIR and other similar research datasets can contribute directly to improved clinical reporting, and hence to better clinical care.

Another potential limitation is the loss of data from women who move outside the catchment area within the index pregnancy and thereafter, although it is our intention to supplement the linkages with data from national HES to provide information on hospitalised outcomes. As with all administrative clinical datasets, research will be limited to that which can be conducted using information routinely collected in clinical care. The intended incorporation of research datasets offsets this to some extent. Additionally, clinical data entry may involve substantial human error and data may be absent from the clinical records (ie, missing). The representativeness of the cohort to the UK population is limited to mixed, inner-urban catchments. In addition, all three NHS Trusts involved in the eLIXIR Partnership incorporate specialisms or local expertise attracting national-level referrals. As a result, data may be skewed to patients with more severe or complex health issues.

There are several advantages, but also disadvantages, to using pseudonymised electronic cohorts versus more traditional consented cohorts (eg, Avon Longitudinal Study of Parents and Children or Born in Bradford (BiB)). The greatest advantage lies in the contemporary reporting of a population compared with historical cohorts, and we are aware that BiB has embarked on a

**Table 2** Characteristics of births and birth outcomes at Guy's & St. Thomas' NHS Foundation trust (GSTT) and King's College Hospital NHS Foundation Trust (KCH) between first October 2018 and 30th June 2019

| | GSTT (n=5697) | KCH (n=2708) | Across both NHS trusts (n=8405) |
|---|---|---|---|
| Gestation at delivery (weeks) | 38.8±2.4 | 38.9±2.3 | 38.8±2.4 |
| Very premature (<34 weeks), n (%) | 193 (3.4) | 91 (3.4) | 284 (3.4) |
| Premature (>34 weeks–<37 weeks), n (%) | 295 (5.2) | 117 (4.3) | 412 (4.9) |
| Term (>37 weeks), n (%) | 5209 (91.4) | 2708 (92.3) | 7709 (91.7) |
| No of infants born, (no of pregnancies), n (%) | | | |
| Singleton | 5459 (95.8) | 2592 (95.7) | 8051 (95.8) |
| Twins | 227 (4.0) | 114 (4.2) | 341 (4.1) |
| Triplets | ND* | ND* | 13 (0.2) |
| Type of delivery, n (%) | | | |
| Breech | 32 (0.6) | 16 (0.6) | 48 (0.6) |
| Elective caesarean section | 862 (15.1) | 412 (15.2) | 1274 (15.2) |
| Emergency or unspecified caesarean section | 1132 (19.9) | 461 (17.0) | 1593 (19.0) |
| Forceps | 462 (8.1) | 180 (6.6) | 642 (7.6) |
| Spontaneous cephalic | 2838 (49.8) | 1378 (50.9) | 4216 (50.2) |
| Ventouse | 371 (6.5) | 257 (9.5) | 628 (7.5) |
| Birth outcome, n (%) | | | |
| Live birth | 5644 (99.1) | 2680 (99.0) | 8324 (99.0) |
| Stillbirth or neonatal death | 53 (0.9) | 28 (1.0) | 81 (1.0) |
| Birth weight (grams) | 3249.1±621.7 | 3273.3±621.25 | 3256.9±621.6 |
| Small for gestational age (birth centile ≤10th), n (%) | 551 (9.7) | 259 (9.6) | 810 (9.7) |
| Large for gestational age (birth centile ≥90th), n (%) | 551 (6.6) | 172 (6.4) | 549 (6.6) |

*Not disclosed (ND). Actual numbers suppressed to reduce risk of statistical disclosure.

**Table 3** Characteristics of patients admitted to neonatal intensive care unit (NICU) at Guy's & St. Thomas' NHS Foundation trust (GSTT) and King's College Hospital NHS Foundation Trust (KCH) between 1 October 2018 and 30 June 2019

| | GSTT (n=645) | KCH (n=302) | Total (n=947) |
|---|---|---|---|
| Gestational age (weeks) | 35.2 (±5.0) | 35.7 (±4.6) | 35.3 (±4.9) |
| Birth weight (grams) | 2541.1 (±1075.1) | 2586.4 (±1022.2) | 2555.6 (±1051.32) |
| No days on neonatal unit | 15.3±24.7 | 16.9 (±26.2) | 15.8 (±25.2) |
| No of episodes of care, n (%) | | | |
| 1 | 476 (73.8) | 254 (84.1) | 730 (77.1) |
| 2 | 135 (20.9) | 39 (12.9) | 174 (18.4) |
| 3 | ND* | ND* | 30 (3.2) |
| 4+ | ND* | ND* | 13 (1.2) |
| Readmission (yes), n (%) | 25 (3.9) | 11 (3.6) | 36 (3.8) |
| Consanguineous parents | ND* | ND* | 11 (1.2) |
| Reason for admission, n (%) | | | |
| Preterm birth | 132 (20.5) | 73 (24.2) | 205 (21.6) |
| Respiratory disease | 172 (26.7) | 90 (29.8) | 262 (27.6) |
| Cardiovascular disease | *ND | *ND | 93 (9.8) |
| Infection | 11 (1.7) | 12 (4.0) | 23 (2.4) |
| Jaundice | 29 (4.5) | 11 (3.6) | 40 (4.2) |
| Poor feeding/weight loss | ND* | ND* | 21 (2.2) |
| Gastrointestinal disease | ND* | ND* | 24 (2.5) |
| Hypoglycaemia | 34 (5.3) | 13 (4.3) | 47 (5.0) |
| Convulsions or neurological disease | ND* | ND* | 12 (1.3) |
| Congenital abnormality | ND* | ND* | 28 (2.3) |
| Surgery | 67 (10.4) | 27 (8.9) | 94 (9.9) |
| Investigation/ monitoring | 26 (4.0) | 14 (4.6) | 40 (4.3) |
| Poor condition at birth | ND* | ND* | 22 (2.3) |
| Other | 19 (2.9) | 17 (5.6) | 36 (3.8) |
| Maternal drug use | ND* | ND* | 13 (1.4) |
| Maternal smoking (n) | 23 (3.6) | 20 (6.6) | 43 (4.5) |
| Neonatal outcome, n (%) | | | |
| Ward | 281 (43.6) | 152 (50.3) | 433 (45.7) |
| Home | 181 (28.1) | 97 (32.1) | 278 (29.4) |
| Died | 27 (4.2) | 13 (4.3) | 40 (4.2) |
| Unknown | 156 (24.2) | 40 (13.2) | 196 (20.7) |

*Not disclosed (ND). Actual numbers suppressed to reduce risk of statistical disclosure.
NHS, National Health Service.

mother–child linkage involving all mothers who consent to provide their pseudonymised routine data and that of their infants (eg, BiB 2019; borninbradford.nhs.uk/what-we-do/pregnancy-early-years/born-in-bradford). This and other planned 'local' UK linkages provide an opportunity for collaboration through meta-analysis to compare and contrast with other diverse UK populations. Another key advantage of pseudonymised records is the

increased representativeness of under-represented groups of patients, often missing from traditional cohorts. Also, this approach avoids the cost implications of consent and cohort maintenance. However, traditional cohorts have

**Table 4** Mental health data (South London and Maudsley NHS Foundation Trust, SLaM) of women receiving antenatal care at Guy's & St. Thomas' NHS Foundation Trust (GSTT) and King's College Hospital (KCH) NHS Foundation Trust between 1 October 2018 and 30 June 2019

| | n (%) | | |
|---|---|---|---|
| | GSTT (n=5700) | KCH (n=4507) | Across both NHS trusts (n=10 207) |
| Ever received treatment from SLaM | 658 (11.5) | 526 (11.7) | 1184 (11.6) |
| Under SLaM Care during pregnancy | 180 (3.2) | 127 (2.8) | 307 (3.0) |
| Received psychological therapy (through IAPT) in pregnancy from SLaM | 113 (2.0) | 88 (2.0) | 201 (2.0) |

IAPT, Improving Access to Psychological Therapy; NHS, National Health Service; SLaM, South London and Maudsley NHS Foundation Trust .

the advantage of much greater depth of biological and psychological information derived from procedures and validated questionnaires.

## Plans for the future

As the eLIXIR database develops, expansions will incorporate health and social care data. The next phase of linkage will comprise a local primary care data resource, Lambeth DataNet (https://selondonccg.nhs.uk/in-your-area/lambeth/our-local-plans/), for which all necessary approvals are in place. Beyond-catchment hospitalisation data could be usefully captured by linkage to national data sources, for example, HES, as mentioned, which has been incorporated in the CRIS platform.[33] Subject to approval, later linkages will incorporate prescribing and education data, in addition to a broader range of local healthcare information as eLIXIR infants enter age ranges covered by other specialties. Finally, the replication of this data-linkage model in other geographical settings in the UK would offer the potential to develop a national data network allowing both larger research cohorts and cross-site replication.

## COLLABORATION

We have established a research database of maternity, neonatal and mental health clinical data not only combining maternal physical and mental health clinical data during pregnancy and later neonatal health, but also providing added value through the potential for the addition of biological measurements (ie, omics data) from the eLIXIR Tissue Bank samples. The eLIXIR Programme has the capacity to continue to grow and develop exponentially, through internal and external collaborations, with small levels of attrition to follow-up and the ability to be used for both common and rare research outcomes. Furthermore, unlike comparable research programmes, the population we sample from is diverse on both ethnicity and sociodemographic levels providing richness of data, which has the potential to lead to exciting research findings.

**Author affiliations**
[1]Department of Psychological Medicine, Institute of Psychiatry Psychology and Neuroscience, King's College London, London, UK
[2]NIHR Maudsley Biomedical Research Centre, South London and Maudsley NHS Foundation Trust, London, UK
[3]Women's College Research Institute, Women's College Hospital, Toronto, Ontario, Canada
[4]Department of Women and Children's Health, King's College London, London, UK
[5]Women's Health Academic Centre KHP, Guy's and Saint Thomas' Hospitals NHS Trust, London, UK
[6]Section of Women's Health, Institute of Psychiatry Psychology and Neuroscience, King's College London, London, UK

**Collaborators** The eLIXIR Partnership.

**Contributors** LEC, LH, RS and LP contributed to the conception of this article. LEC, BA, AJ, CLT, AF, CG, LH, RS and LP were involved in manuscript writing and revision. LEC, MB and LP were involved in data analysis and interpretation. All authors read and approved the final manuscript.

**Funding** This work was supported by an MRC Partnership Grant [MR/P003060/1]. LC and LH receives salary support from the National Institute for Health Research (NIHR) Applied Research Collaboration South London (NIHR ARC South London) at King's College Hospital NHS Foundation Trust. LC was additionally supported by a Medical Research Council Mental Health Data Pathfinder Award to King's College London. LC, AJ, MB, LH, RS are part-funded by the National Institute for Health Research (NIHR) Biomedical Research Centre at the South London and Maudsley NHS Foundation Trust and King's College London. AF and LP are supported by Tommy's Charity and King's College London. CG and LP are supported by the National Institute for Health Research (NIHR) Biomedical Research Centre at Guy's and St. Thomas' NHS Foundation Trust.

**Disclaimer** The views expressed are those of the authors and not necessarily those of the NHS, the NIHR, the MRC or the Department of Health.

**Competing interests** RS declares research support received in the last 5 years from Roche, Janssen, GSK and Takeda.

**Patient and public involvement** Patients and/or the public were involved in the design, or conduct, or reporting, or dissemination plans of this research.

**Patient consent for publication** Not required.

**Ethics approval** Oxford C REC (18/SC/0086) and Cambridge East REC (18/EE/0120).

**Provenance and peer review** Not commissioned; externally peer reviewed.

**Data availability statement** Data are available upon reasonable request. Researchers can apply for data access and biomaterial by submitting a research application form to the eLIXIR team. The eLIXIR website provides information on the application process (http://www.guysandstthomasbrc.nihr.ac.uk/microsites/elixir). To apply to use data from eLIXIR, researchers must complete a Research Application Form (RAF), available on our website, and submit this, via email, to the eLIXIR Oversight Committee for their consideration and approval. The associated costs with accessing data are study dependent. Basic infrastructure for data storage and CDLS services is provided by the core team. Individual project costs are determined by the length of study and which datasets are required. Costs to the researcher include data access (via VPN), data cleaning and statistical support. The eLIXIR Partnership provides the infrastructure for data linkage, but external funding will be sought for additional linkage to external datasets.

**ORCID iDs**
Lauren E Carson http://orcid.org/0000-0002-7027-3077
Amelia Jewell http://orcid.org/0000-0002-0887-2159

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
