## [Reviewer comments · BMJ Open]

ARTICLE DETAILS

TITLE (PROVISIONAL)	Cohort Profile: The eLIXIR Partnership - a maternity-child data linkage for Life Course Research in South London, UK.
AUTHORS	Carson, Lauren; Azmi, Borscha; Jewell, Amelia; Taylor, Clare; Flynn, Angela; Gill, Carolyn; Broadbent, Matthew; Howard, Louise; Stewart, Robert; Poston, Lucilla

VERSION 1 – REVIEW

REVIEWER	Professor Lorna Fraser University of York United Kingdom
REVIEW RETURNED	18-Jun-2020

GENERAL COMMENTS	The authors present a cohort profile for the Elixir mother -child dataset. The ability to utilise routinely collected data for research and public health purposes is important. Major changes  1. the authors state that this is a population based cohort rather than hospital based cohort. More information on clear geographical boundaries and the underlying population that lives in this area is required to evidence this. a comparison between the underlying population data and the demographic data from the cohort would also be useful. 2. the authors state that these data are anonymised, are they not pseudonymised? 3. it is good to see that the authors have focused on the legal, governance and ethical frameworks in which the Elixir cohort is based. I think it would be helpful if this section was clearer about the approvals they needed to process and link these data and the processes for researchers who wish to access these data. 4. there are other ongoing similar electronic cohorts e.g. Born in Bradford for all https://borninbradford.nhs.uk/what-we-do/pregnancy-early-years/born-in-bradford/. the ADRN in London were also working on national electronic birth cohorts. 5. It would also be good to see some more comments on other available dataset such as CPRD, mother baby linkages. 6. Some more comment on the advantages and disadvantages of full population electronic cohorts versus the more traditional birth cohorts e.g. ALSPAC, BiB etc
--

REVIEWER	Alicia Montgomerie The University of Adelaide, Australia
REVIEW RETURNED	03-Jul-2020

GENERAL COMMENTS	In this cohort profile the authors describe The Early Life Cross Linkage in Research (eLIXIR) Partnership, which they “aim to
---

combine maternal, infant and child health data into a single resource to allow information from large numbers of mothers, babies and children to be investigated over an unlimited time period” to “support investigations into associations between physical and mental health in mother and child” and to also to include the eLIXIR Research Tissue Bank to link the routinely collected maternal and neonatal clinical data with biological samples.

Comments

1. Page 6 line 7 please write out ‘eLIXIR’ as this is the first in-text reference to the acronym.
2. Line 3 page 6 the authors stated ‘To our knowledge, no clinical data linkages in maternity or neonatal services have to date incorporated both physical and mental health data to interrogate these associations at a population level’ and Line 27 page 7 ‘We believe this is one of the first longitudinal research programmes from early pregnancy onward using routinely collected clinical data which does not rely on a recruited cohort of participants’. There are a number of linkage projects in Europe, Scotland, Australia, US and more that have linked linkage clinical data from with published papers that detail this, including The Aberdeen Maternity and Neonatal Databank (AMND). Please re-word these sentences to be more specific about what clinical data will be linked as a number of studies link such neonatal pregnancy outcomes and episodes.
3. Page 7 line 45 (Maternity, birth and neonatal intensive care data). Please provide an examples of the type of data available in the maternity, birth and neonatal intensive care data (e.g. anthropometric, hospital episode) to provide readers with a clear picture of the data available and ideas of potential research questions that can be answered.
4. Page 8 line 3 (Mental health data). Please provide an examples of the type of data available in the Mental health data (e.g. diagnoses, services provided,) to provide readers with a clear picture of the data available.
5. Page 9 line 25 (Data Linkage procedures and resources). The authors state deterministic matching techniques was used. Please provide information on the quality of the linkage (matching of records) for example the match rate and link accuracy. Then readers can understand the quality of the linkage for this project.
6. Page 9 line 40: the authors stated that Public and patient involvement (PPI) is ensured in the decision-making process of approving projects using linked data held by eLIXIR. For the readers to understand please state how public and patient involvement will be undertaken?
7. Page 9 line 49 (Research Tissue Bank). The authors stated for inclusion into the research tissue bank the women give informed consent. It would be useful for readers to understand the number of women that give consent. Please provide initial rates of consent for inclusion into the research tissue bank.
8. Page 10 line 3 (Maternal blood sample collection). Authors stated informed consent will be obtained for the maternal blood sample collection. It would be useful for the readers to understand the number of women that gave consent. Please provide initial rates of

	consent. 9. Page 11 line 13 (Infant blood sample collection). For infants blood sample collection informed parental consent is obtained. It would be useful for readers to understand the numbers that give consent. Please provide initial rates of consent. 10. Page 17 line 30 (Plans for the future). The authors state there is current approval to incorporate health and social care data and other additional data to be incorporate subject to approvals. Add data to projects takes time and is costly, is the project currently have funding to add these datasets or while further funding be sort? 11. Page 18 line 15 (Data sharing statement). Authors state that researchers can apply for data access. Please provide readers if there will be a cost associated with accessing the data and obtaining future data linkages. Please elaborate on any 'No' answers in the free text section below. 3. Is the study design appropriate to answer the research question? N/A – The authors have not stated any specific research questions in the paper so unable to ascertain if the study design is appropriate. 6. Are the outcomes clearly defined? N/A – The authors have not stated any specific research questions in the paper so unable to ascertain if the study design is appropriate.
--	--

VERSION 1 – AUTHOR RESPONSE

Reviewers' Comments to Author:

Reviewer: 1

Point 1: the authors state that this is a population-based cohort rather than hospital based cohort. More information on clear geographical boundaries and the underlying population that lives in this area is required to evidence this. a comparison between the underlying population data and the demographic data from the cohort would also be useful.

We have now included the geographical areas which GSTT, KCH and SLAM serve.

Page 6, line 23-30; GSTT provide a full range of hospital and community services for people in Lambeth, Southwark and Lewisham, as well as specialist care for patients from further afield including referrals for high risk pregnancies and neonatal complications. Similarly, KCH serves the boroughs of Lambeth, Southwark, and Lewisham, but also Bromley, with specialist services to patients across a wider catchment area, including referrals for obstetrics and fetal medicine. SLAM provides comprehensive mental health services to a geographic catchment of over 1.2 million residents in four south London boroughs, Croydon, Lambeth, Lewisham and Southwark, as well as some regional/national specialist mental health services.

Point 2: the authors state that these data are anonymised, are they not pseudonymised?

We apologise for this oversight. We have amended the manuscript so that the data is referred to as de-identified rather than anonymous.

Point 3: it is good to see that the authors have focused on the legal, governance and ethical frameworks in which the Elixir cohort is based. I think it would be helpful if this section was clearer about the approvals they needed to process and link these data and the processes for researchers who wish to access these data.

Considerable detail was originally provided in the manuscript but we have added the following text to improve the clarity around the approvals needed to process, link and access data within eLIXIR. Page 9, line 19-21; Therefore, for current and future data linkages within eLIXIR, ethical (REC) and s251 approval is required either through amendments to our existing agreements or new

applications to these regulatory bodies.

Page 17, line 4-16; To apply to use data from eLIXIR, researchers must complete a Research Application Form (RAF), available on the study website, and submit this, via email, to the eLIXIR Oversight Committee for their consideration and approval.

Point 4: there are other ongoing similar electronic cohorts e.g. Born in Bradford for all <https://borninbradford.nhs.uk/what-we-do/pregnancy-early-years/born-in-bradford/>. the ADRN in London were also working on national electronic birth cohorts. It would also be good to see some more comments on other available dataset such as CPRD, mother baby linkages.

We have added reference to the CPRD and ADR UK linkages to the list of existing linkages and changed the text in the introduction as below.

Page 4, line 26-30; Several linkages of clinical maternity and infant data have nonetheless shown the feasibility of the approach, and usefulness, for example in aligning hospital maternity data with national birth registration datasets, or birth registration datasets with Hospital Episode Statistics , or using UK primary care pregnancy data to create a pregnancy register 15-25.

New References:

24. Minassian C, Williams R, Meeraus WH, et al. Methods to generate and validate a Pregnancy Register in the UK Clinical Practice Research Datalink primary care database.

Pharmacoepidemiology

and Drug Safety 2019;28(7):923-33. doi: 10.1002/pds.4811

25. ADR UK. Birth cohort data linkage study [Available from: <https://www.adruk.org/ourmission/our-impact/article/birth-cohort-data-linkage-study-228/>].

We are also aware that Born in Bradford has recently initiated routine maternity data collection for the purpose of linkage to child health data (we have indeed discussed with Prof John Wright, lead for BiB) but this has yet to report a protocol.

In recognition of this new (BiB2019) study, we have now included a sentence in the discussion to refer to this new data-linkage study and to emphasise the potential for collaboration.

Page 15, line 33 – page 16 line 4; ... and we are aware that Born in Bradford has embarked on a mother-child linkage involving all mothers who consent to provide their de-identified routine data, and that of their infants (e.g. Born in Bradford 2019; borninbradford.nhs.uk/what-we-do/pregnancy-early-years/born-in-bradford). This and other planned 'local' UK linkages provide opportunity for collaboration through meta-analysis to compare and contrast with other diverse UK populations.

Point 5: Some more comment on the advantages and disadvantages of full population electronic cohorts versus the more traditional birth cohorts e.g. ALSPAC, BiB etc

We have amended the manuscript to address this point with the following paragraph in the discussion.

Page 15, line 30 – page 16, line 8; There are several advantages, but also disadvantages to using deidentified

electronic cohorts versus more traditional consented cohorts (e.g. Avon Longitudinal Study of Parents and Children (ALSPAC) or Born in Bradford (BiB)). The greatest advantage lies in the contemporary reporting of a population compared with historical cohorts, and we are aware that Born in Bradford has embarked on a mother-child linkage involving all mothers who consent to provide their de-identified routine data, and that of their infants (e.g. Born in Bradford 2019; borninbradford.nhs.uk/what-we-do/pregnancy-early-years/born-in-bradford). This and other planned 'local' UK linkages provide opportunity for collaboration through meta-analysis to compare and contrast with other diverse UK populations. Another key advantage of de-identified records is the increased representativeness of underrepresented groups of patients, often missing from traditional cohorts. Also, this approach avoids the cost implications of consent and cohort maintenance. However, traditional cohorts have the advantage of much greater depth of biological and psychological information derived from procedures and validated questionnaires.

Reviewer: 2

Point 1: Page 6 line 7 please write out 'eLIXIR' as this is the first in-text reference to the acronym. We have amended this sentence to ensure that 'eLIXIR' is written out in full when first referenced in the main body of the text.

Page 5, line 2-4; "The Early Life Cross Linkage in Research (eLIXIR) Partnership has been developed to

address these relationships from early pregnancy, the perinatal period and beyond into later life."

Point 2: Line 3 page 6 the authors stated 'To our knowledge, no clinical data linkages in maternity or neonatal services have to date incorporated both physical and mental health data to interrogate these associations at a population level' and Line 27 page 7 'We believe this is one of the first longitudinal research programmes from early pregnancy onward using routinely collected clinical data which does not rely on a recruited cohort of participants'. There are a number of linkage projects in Europe, Scotland, Australia, US and more that have linked linkage clinical data from with published papers that detail this, including The Aberdeen Maternity and Neonatal Databank (AMND). Please re-word these sentences to be more specific about what clinical data will be linked as a number of studies link such neonatal pregnancy outcomes and episodes.

We have again searched the literature and can find no linkages which reflect a combination of data from maternal, neonatal and mental health records. This we believe is the unique aspect of eLIXIR. We have re-written these sentences, in the introduction and discussion, to clarify that we believe that eLIXIR is the first data linkage programme which has linked clinical data from maternity, neonatal and psychiatric services into a single continuum.

Page 4, line 34 – page 5, line 2; To our knowledge, no clinical data linkages in maternity or neonatal services have to date incorporated clinical information from maternity, neonatal and mental health services into a single continuum to interrogate these associations at a population level.

Page 6, line 15-18; eLIXIR is one of the first longitudinal research databases which from early pregnancy onwards using routinely collected clinical data from maternity, neonatal and mental health services that does not rely on a recruited cohort of participants.

Point 3: Page 7 line 45 (Maternity, birth and neonatal intensive care data). Please provide an examples of the type of data available in the maternity, birth and neonatal intensive care data (e.g. anthropometric, hospital episode) to provide readers with a clear picture of the data available and ideas of potential research questions that can be answered.

We have now incorporated the type of data as stated below:

Page 7, line 3-5; Within GSTT and KCH the BadgerNet platforms are used for recording maternal/infant personal data, demographics, clinical history, clinic data (maternity only) and hospital episode data.

Point 4: Page 8 line 3 (Mental health data). Please provide an examples of the type of data available in the Mental health data (e.g. diagnoses, services provided,) to provide readers with a clear picture of the data available.

We have now incorporated the type of mental health data available as stated below:

Page 7, line 21-24; In terms of cohort coverage, all SLAM care (including diagnoses, medication and services provided), is represented on CRIS, including Improving Access to Psychological Therapies data (IAPT; a large primary care service providing short term psychological therapies).

Point 5: Page 9 line 25 (Data Linkage procedures and resources). The authors state deterministic matching techniques was used. Please provide information on the quality of the linkage (matching of records) for example the match rate and link accuracy. Then readers can understand the quality of the linkage for this project.

As we are not combining data from datasets where patients may or may not be present (e.g. mental health services), we are not able to provide a match rate. However, we have added the following statement to the manuscript to state how effective our matching is between mother and infant pairings.

Page 9, line 4-5; Match quality is assured as 100% of infants born within the eLIXIR dataset were matched with their mothers' records with the BadgerNet System.

Point 6: Page 9 line 40: the authors stated that Public and patient involvement (PPI) is ensured in the

decision-making process of approving projects using linked data held by eLIXIR. For the readers to understand please state how public and patient involvement will be undertaken?

This sentence has been re-written to make it clearer what the PPI involvement is within eLIXIR, and we have expanded this section to include information about PPI throughout the development of the linkage.

Page 9, line 33 – page 10 line 5; Public and patient involvement (PPI) involvement was incorporated throughout the development of the eLIXIR Partnership. The concept of the eLIXIR Partnership was presented to a variety of PPI groups, including the Maudsley Biomedical Research Centre (BRC) Data

Linkage Service User and Carer Advisory Group³⁷, Lambeth HealthWatch and the Young Persons Advisory Group at Great Ormond Street Hospital. PPI is ensured in the decision-making process of approving eLIXIR projects through lay member representation on the eLIXIR Oversight Committee. The eLIXIR Oversight Committee reviews and approves all projects using eLIXIR data.

Point 7: Page 9 line 49 (Research Tissue Bank). The authors stated for inclusion into the research tissue bank the women give informed consent. It would be useful for readers to understand the number of women that give consent. Please provide initial rates of consent for inclusion into the research tissue bank.

Please see responses to Point 8 and Point 9.

Point 8: Page 10 line 3 (Maternal blood sample collection). Authors stated informed consent will be obtained for the maternal blood sample collection. It would be useful for the readers to understand the number of women that gave consent. Please provide initial rates of consent.

We have now included the rates of consent for maternal blood sample collection.

Page 14 line 15 – page 15 line 2; Following all necessary governance agreements samples were collected over a period of 3 months. A total of 1271 aliquoted samples (including EDTA, serum and whole blood) from 123 women were stored in the FreezerPro© system. In this period, 63.4% of women approached gave consent to take part.

Point 9: Page 11 line 13 (Infant blood sample collection). For infants blood sample collection informed parental consent is obtained. It would be useful for readers to understand the numbers that give consent. Please provide initial rates of consent.

During the reporting period of this manuscript (October 2018-June 2019) the NICI sample collection had not been established. However, recruitment is now underway. The following paragraph has been amended under the research tissue bank section.

Page 10, line 22-25; It is our intention also to recruit samples from infants admitted to the neonatal intensive care unit (NICU), when blood is drawn for routine tests. Following written informed parental consent, residual blood from routine samples, which otherwise would be discarded, will be retained and collected. The samples will be processed and stored in a similar manner to the maternal blood samples.

Point 10: Page 17 line 30 (Plans for the future). The authors state there is current approval to incorporate health and social care data and other additional data to be incorporate subject to approvals. Add data to projects takes time and is costly, is the project currently have funding to add these datasets or while further funding be sort?

We can confirm that there is funding in place to support the linkages listed within the manuscript.

Page 17, line 10-11; The eLIXIR Partnership provides the infrastructure for data linkage, but external funding will be sought for additional linkage to external datasets.

Point 11: Page 18 line 15 (Data sharing statement). Authors state that researchers can apply for data access. Please provide readers if there will be a cost associated with accessing the data and obtaining future data linkages.

We have now amended this sentence to state the current associated costs with accessing the data through the eLIXIR platform. We are unable to comment on the costs of future data linkages as they will be dependent on the external parties providing the data.

Page 17, line 6-10; The associated costs with accessing data are study dependent. Basic infrastructure for data storage and CDLS services is provided by the core team. Individual project

costs are determined by the length of study and which datasets are required. Costs to the researcher include data access (via VPN), data cleaning and statistical support.

VERSION 2 – REVIEW

REVIEWER	Prof Lorna Fraser University of York
REVIEW RETURNED	29-Jul-2020

GENERAL COMMENTS	The authors have addressed most of my comments. However I would prefer that they replace the terminology of de-identified to pseudonymised. This is important as within the GDPR pseudonymised data is still personal data see https://ico.org.uk/for-organisations/guide-to-data-protection/guide-to-the-general-data-protection-regulation-gdpr/what-is-personal-data/what-is-personal-data/
--

REVIEWER	Alicia Montgomerie The University of Adelaide, Australia
REVIEW RETURNED	14-Aug-2020

GENERAL COMMENTS	The authors have address all questions from previous review. I have no further comments.
--